# Recent Advances in Layered-Double-Hydroxides Based Noble Metal Nanoparticles Efficient Electrocatalysts

**DOI:** 10.3390/nano11102644

**Published:** 2021-10-08

**Authors:** Zexuan Zhang, Peilong Li, Xin Zhang, Cun Hu, Yuwen Li, Bin Yu, Ning Zeng, Chao Lv, Jiangfeng Song, Mingcan Li

**Affiliations:** 1Xinjiang Key Laboratory of Solid State Physics and Devices, School of Physical Science and Technology, Xinjiang University, Urumqi 830046, China; zhangzexuan2018@163.com; 2Institute of Materials, China Academy of Engineering Physics, Jiangyou 621908, China; a753951_xin@163.com (X.Z.); hucun402@163.com (C.H.); limliayuw@gmail.com (Y.L.); fusionchina@126.com (B.Y.); dtbanxia@126.com (N.Z.); lvchao219@foxmail.com (C.L.); 3School of Nuclear Science and Engineering, East China University of Technology, Nanchang 330013, China

**Keywords:** LDHs-based catalysts, noble metal catalysts, composite materials, hydrogen evolution reaction, oxygen evolution reaction

## Abstract

With the energy crisis and environmental pollution becoming more and more serious, it is urgent to develop renewable and clean energy. Hydrogen production from electrolyzed water is of great significance to solve the energy crisis and environmental problems in the future. Recently, layered double hydroxides (LDHs) materials have been widely studied in the electrocatalysis field, due to their unique layered structure, tunable metal species and highly dispersed active sites. Moreover, the LDHs supporting noble metal catalysts obtained through the topotactic transformation of LDHs precursors significantly reduce the energy barrier of electrolyzing water, showing remarkable catalytic activity, good conductivity and excellent durability. In this review, we give an overview of recent advances on LDHs supporting noble metal catalysts, from a brief introduction, to their preparation and modification methods, to an overview of their application in the electrocatalysis field, as well as the challenges and outlooks in this promising field on the basis of current development.

## 1. Introduction

As the world’s population grows, the demand of energy will increase rapidly in the foreseeable future [1]. The vast majority of modern energy comes from coal, oil, natural gas or other fossil fuels, which has caused a serious energy crisis and environmental problems [2]. Recently, growing concerns about serious environmental problems and the energy crisis have urged us to seek sustainable energy as a viable alternative to traditional fossil fuels [3]. Hydrogen as a high-energy-density carrier is considered to be a clean and renewable alternative to fossil fuels and plays a vital role in promoting the sustainable energy development of human society [4,5]. Electrocatalytic water is an effective renewable energy production method for converting electric energy into chemical energy stored in hydrogen fuel. Compared with the traditional high-temperature and high-pressure steam conversion of fossil fuels, this reaction has advantages of high purity, good environmental conditions and low energy consumption [6,7]. In recent years, the design and control of synthetic electrocatalysts has played an active role in the development of electrochemistry and catalysts [8]. To reduce the loss of charge transfer in electrochemical processes, traditional water electrolysis is usually carried out in acidic conditions using a proton exchange membrane or in alkaline media using a diaphragm (Figure 1a). Hydrogen evolution reaction (HER) and oxygen evolution reaction (OER) are two semi-reactions of electrolyzed water, both of which are critical to overall efficiency (Figure 1b). Theoretically, it would take 1.23 V vs. normal hydrogen electrode (NHE) to electrolyze water [9]. Unfortunately, the slow kinetics of HER and OER lead to lower energy efficiency [10,11]. Therefore, the input potential of cracked water in the actual electrolyzer is much higher than 1.23 V [12,13]. The large overpotential and slow kinetics of HER and OER hinder the practical application of whole-water decomposition [14,15]. Therefore, the design of electrocatalysts with high electrocatalytic activity is of great importance to reduce the overpotential and improve the efficiency of water decomposition.

Generally, precious metals (such as Pt) and precious metal oxides (like RuO_2_ and IrO_2_) are considered to be excellent HER and OER electrocatalysts, respectively [17,18]. However, due to their limited reserves and high prices, it is too costly for widespread commercial application [19]. Thus, researchers have begun to explore effective and efficient strategies to achieve the high dispersion of active substances, so as to optimize the use of noble metal resources [15]. The study found that the tunability of the microstructures and electronic states of active components on the atomic scale can provide higher performance for electrochemical reactions [20,21]. Unfortunately, it is still a difficult and challenging problem to improve the utilization rate of active components, the rate of catalytic reaction and the thermal stability of the catalyst by controlling the dispersion of active sites and the microstructure [22]. Loading the catalyst on suitable supporters (such as carbon materials [23,24,25], metals [26,27], metal oxide [28,29], metal hydroxide [30,31], metal-organic frameworks [32,33,34] and boron nitride [35,36]) is an optimal strategy to solve this problem. In particular, layered double hydroxides (LDHs) and their derivatives (metal hydroxides, hydroxides, oxides, sulfides, nitrides and phosphides) are widely studied in water splitting, due to their compatible properties with traditional noble metals, easy synthesis, low cost, rich resources, good activity and long-time durability [37,38]. The two or more kinds of metal cations in the LDHs plate are uniformly dispersed on the atomic scale, and the metal cations do not agglomerate [39]. Each single metal cation with catalytic activity can be used as a single catalytic active site, leading to high catalytic activity in the electrocatalysis process [40]. When used as supporters, the higher specific surface areas of LDHs can not only effectively prevent the agglomeration and deactivation of catalyst nanoparticles but can also play the role of anchoring the catalyst and adjusting its shape and size [41]. In recent years, some important reviews on the development of LDHs-based materials used as electrolyzed water catalysts have been summarized [42,43]. However, these early reviews mainly focused on synthesis methods and techniques, such as exfoliation techniques, interlayer techniques, etc. The reports of LDHs supporting noble metal nanoparticles are less concerning, which is considered to be of great significance for the design of sustainable energy materials [44,45]. In this review, the research progress of LDHs supporting noble metal catalysts in recent years is reviewed (Figure 1). The research on the intelligent design and synthesis of LDHs supporting noble metal catalysts in the field of water electrolysis is briefly introduced from the aspects of preparation methods and modification methods. The advantages of LDHs as catalysts and catalyst carriers are emphasized, and the relationship between structures and properties is discussed. Combined with the current research progress, the challenges and prospects in this field are put forward. This provides a useful insight for developing stable and promising catalysts and catalyst carriers based on LDHs.

## 2. LDHs-Based Electrocatalysts

LDHs are a class of typical anionic layered materials with a structure similar to that of magnesium hydroxide (Figure 2). The chemical formula is usually [M^2+^_1−X_M^3+^_X_ (OH)_2_]^x+^(A^n−^)_x/n_·mH_2_O, where M^2+^ and M^3+^ are metal cations located in the main plate layer, An- is the interlaminar anion, and m is the number of interlaminar crystal water in LDHs [46,47]. Structurally, each bivalent cation is surrounded by six hydroxyl octahedra; sharing edges; and forms an infinite sheet metal layer [48,49]. The metals are stacked together to form a layered structure held together by hydrogen bonds or Van der Waals force [50]. Many metal ions can be used to synthesize LDHs with stable structures, such as divalent metal cations Mg^2+^, Fe^2+^, Co^2+^, Ni^2+^, Cu^2+^, Zn^2+^, Mn^2+^ and Cd^2+^, and trivalent metal cations Al^3+^, Co^3+^, Fe^3+^, Mn^3+^, Rh^3+^, Ru^3+^, Cr^3+^ and V^3+^ [46]. Due to the extensive adjustability of the metal cations of LDHs, the exchangeability of the interlayer bound anions, and the large number of physical and chemical properties assemblies, nanostructures can be rationally designed [51,52]. Li et al. [53] used a simple one-step hydrothermal method to grow Gd-doped NiFe-LDH in situ. Gd doping optimizes the electronic structure of NiFe-LDH, increases the number of its oxygen vacancies, enriches its layered porous morphology and makes it have excellent OER electrocatalytic activity in alkaline medium. Furthermore, LDHs can be used as support and precursors for different catalysts [38,54]. Due to the diversity of their types and compositions, as well as the different proportions and arrangements of metal ions, it is possible to design different catalysts at the atomic scale [55,56]. Many different kinds of LDHs contain Fe^2+^/Fe^3+^, Co^2+^/Co^3+^, Ni^2+^/Ni^4+^ and Mn^2+^/Mn^3+^ and can be used as active catalysts directly [57]. Wang et al. [58] designed the three-dimensional hollow structure of NiFe-LDH on the surface of NiFe foam (NiFe-LDH@NiFe) by using the acid corrosion induction strategy. It should be noted that the NiFe-LDH@NiFe needs only ultra-low overpotential of 201 mV in 1 M KOH electrolyte to provide 10 mA cm^−2^ current density with excellent stability. The preparation method of LDHs electrocatalyst is simple and easy to obtain, which makes the structure of LDHs easy to be controlled by chemical synthesis [59]. In addition, LDHs have an average specific surface area and electrolytic water activity because of their unique lamellar structure and electronic structure [60,61]. Recently, Wang et al. [62] have used NiCo-LDHs as a carrier to load Pt nanoparticles. Highly dispersed and ultrafine Pt NPs/LDHs-supported catalysts were prepared by using the anchoring effect of LDHs to regulate Pt nanoparticles. This shows high catalytic activity and cycle stability in the electrocatalytic oxidation of methanol.

Recently, various studies on LDHs-based materials as catalysts for electrolyzing water have been published with rapid growth trending. This literature pointed out that the limited number of active sites, the poor conductivity and the poor intrinsic catalytic activity of active sites are the key factors limiting the performance of electrocatalysts [64,65]. Therefore, in order to improve the electrocatalytic activity of LDHs, several strategies have been developed. Increasing the number of active sites is a strategy to improve the catalytic activity of LDHs electrocatalysts [62,66]. Chen et al. [67] found that ultra-thin or monolayer LDHs can be directly synthesized by suitable two-dimensional region-limited growth in solution, exposing more active sites and reducing the over-potential of electrocatalytic reactions. The intrinsic catalytic activity of the active site is a greatly important factor affecting the performance of LDHs electrocatalysts [68,69]. By loading noble metals (such as Pt, Ir, and Ru) onto LDHs with large specific surface areas, the electronic distribution of adjacent atoms can be effectively adjusted; the intrinsic catalytic activity of active sites can be further regulated and improved, and it can also minimize the consume of precious metals and improve the electrocatalytic performance in low cost [70,71]. LDHs can be used as excellent catalyst supports due to their highly specific surface area. Their special lamellar structure and satisfactory properties makes LDHs have highly specific surface area, which can disperse noble metal nanoparticles evenly. It can also play the role of fixing the catalyst and adjusting its shape and size, so that the catalytic active center can be highly dispersed on the LDHs support. Experiments showed that LDHs as a support exhibited higher catalytic activity and cycle stability in electrocatalytic reaction [46,71,72].

## 3. Electrocatalysts of LDHs Supporting Noble Metals

Noble-metal-based catalysts (such as Pt, Ir and Ru) are considered to be electrocatalysts with excellent catalytic activity [73]. Loading noble metal nanoparticles on the surface of LDHs could expose more active sites, reduce the resource utilization and cost, and improve electrocatalytic performance [37,74]. This has been corroborated by many recent reports.

### 3.1. Electrocatalysts of LDHs Supporting Pt

It is well known that Pt has excellent electrocatalytic properties in the reaction of HER, but its high price, scarce reserves and catalyst deactivation limit its application [75]. However, the current Pt is still the best metal catalyst for electrocatalytic hydrogen activity. In order to improve the utilization rate of Pt and reduce the cost of precious metals in catalyst preparation, researchers have been committed to the development of catalysts that can replace Pt in HER applications (such as transition metals Mo, W, etc.) [76]. Recently, Feng et al. [77] electrodeposited Pt nanoparticles on amorphous NiFe-LDH to regulate the interaction between Pt and NiFe-LDH. The experimental results show that the Pt nanoparticles on the non-static NiFe-LDH surface are smaller than those on the crystalline NiFe-LDH surface, and that the size distribution is narrow and uniform. By using the porous structure and highly specific surface area of LDHs, the particle size and dispersibility of Pt can be controlled, and the agglomeration of Pt nanoparticles can be prevented effectively, thus achieving high utilization ratio of Pt [78].

Anantharaj et al. [79] reported that NiFe-LDH crystal plates with high crystallinity were prepared by hydrothermal method, and NiFe-LDH-containing Pt nanoparticles were synthesized by a two-step reaction. It was found that the Pt NPs can be uniformly distributed on the NiFe-LDH crystal prepared by hydrothermal method, but the NiFe-LDH prepared by co-precipitation method cannot be realized (Figure 3a–f). The average size of Pt nanoparticles on the NiFe-LDH crystal plate prepared by hydrothermal method is 4 ± 1 nm (Figure 3a–c). The overpotential and Tafel slope of NiFe-LDH prepared by hydrothermal method with Pt nanoparticles are 27 mV and 51 mV dec^−^^1^, which are lower than NiFe-LDH co-deposited with Pt nanoparticles. These encouraging findings confirm the applicability of high crystallinity Pt nanoparticles supported NiFe-LDH in all-water electrocatalysis. Meanwhile, the unexpected formation of Ni_0.6_Fe_2.4_O_4_ on the surface of NiFe-LDH wafers prepared by hydrothermal method increased the conductivity of NiFe-LDH, thus enhancing the OER and HER activities of NiFe-LDH phase.

One of the most active commercial catalysts (Com-Pt/C) currently widely used for HER is electrically conductive carbon loaded with Pt nanoparticles (20 wt% Pt/C) [80]. However, the content of 20% Pt in Com-Pt/C catalyst was still very high, so the content of Pt should be further reduced. Recently, Yan et al. [81] used self-supporting 3D NiFe LDH on carbon fiber cloth (NiFe LDH/CC) as the starting material and loaded ultrafine Pt sub-nanoparticles on two-dimensional (2D) NiFe LDH nanosheets by chemical reduction to prepare efficient NiFe LDH/CC electrocatalyst (Pt-NiFe LDH/CC) with low Pt content (1.56 wt%). The results showed that strong interaction was formed between Pt sub-nano clusters and 2D NiFe LDH nanosheets, which effectively prevented the aggregation of Pt sub-nano clusters. The Pt sub-nano clusters with average size of 0.59 nm are highly dispersed on the surface of NiFe LDH nanosheets, which can expose more active centers, shorten the electron transfer path and greatly reduce the consume amount of Pt (Figure 4a). The Pt-NiFe LDH/CC electrode has an ultra-low Pt content of 1.56 wt%. The overpotential at a current density of 10 mA cm^−2^ is 28 mV, which is equivalent to the Com-Pt/CC electrode (Figure 4b). After 1000 cycles, the Pt-NiFe LDH/CC electrode showed the same performance as the original measurement; the current density kept 36,000 s at 10 mA cm^−2,^ and the potential was stable, which indicated excellent durability (Figure 4c,d). The Pt-NiFe LDH/CC have excellent electrocatalytic properties, and the design of novel electrocatalysts provides an effective strategy for the structure regulation of the catalysts in the future, in order to achieve efficiency.

### 3.2. Electrocatalysts of LDHs Supporting Ir

Due to the advantages of high catalytic activity, low energy consumption and stability in the OER reaction, iridium-based catalysts have been widely used in electrode materials [82,83]. However, similarly to Pt, Ir has few reserves and is expensive, limiting its large-scale application [84,85]. Therefore, in order to improve the efficiencies of iridium-containing catalysts, new iridium-containing catalysts need to meet the following two points: (1) high catalytic activity with low overpotential and (2) low cost of Ir [86,87]. To this end, the majority of researchers have sought a new catalyst to explore and research. They found that doping noble metal Ir nanoparticles into LDHs not only reduced the dosage of noble metal Ir but also improved its catalytic efficiency [24,88]. In particular, attractive nickel-based layered hydroxides (such as NiFe and NiCo LDHs) exhibit good OER activity during alkaline electrolyzing water. Due to their abundant resources, highly specific surface area and controllable chemical composition, they have attracted extensive attention [89,90].

Recently, some researchers have developed bifunctional electrocatalysts by coupling Ir with LDH. Li et al. [91] constructed Ir-O-V catalytic group by introducing Ir atom as dopant and combining it with substrate material NiV-LDH, in which Ir atom helps to dissociate water molecules and regulate the adsorption energy of bridging oxygen and V atom. The OER and HER overpotentials of the synthesized NiVIr-LDH were 203 mV@10 mA cm^−2^ (Figure 5a) and 42 mV@10 mA cm^−2^ (Figure 5c), respectively. The corresponding tafel slopes are 55.3 mV dec^−1^ (Figure 5b) and 35.9 mV dec^−1^ (Figure 5d). Thus, the total water splitting current of 10 mA cm^−2^ was achieved at a voltage of 1.49 V (as anode and cathode) by the newly prepared NiVIr-LDH catalyst, which is lower than that of the Pt/C and Ir/C coupling (1.60 V@10 mA cm^−2^) (Figure 5e). Compared with other bifunctional electrocatalysts, the electrocatalytic performance of the NiVIr-LDH catalyst is better (Figure 5f). The appropriate binding of noble metal Ir nanoparticles into LDHs is an effective strategy to improve catalytic performance, which can increase catalytic efficiency and reduce the consumption of noble metal Ir; moreover, the new catalytic group has higher performance.

Fan et al. [92] reports a bifunctional catalyst that in situ synthesizes Ir-NiCo LDH over nickel foam (NF) by a two-step process (Figure 6a). The pristine NiCo LDH was first grown on NF by solvothermal method at a low temperature of 90 °C and an atmosphere. Then, the NiCo LDH/NF samples were immersed in the homogeneous precursor solution containing IrCl_3_·3H_2_O and the spontaneous displacement was performed. The results showed that LDHs were composed of many vertically aligned and interconnected nanosheets. (Figure 6b,c). The single Ir atom is highly dispersed in Ir-NiCo LDH/NF catalyst, and the particle size is 0.2 nm (Figure 6d,e). The prepared NiCo-Ir LDH showed excellent catalytic activity. The overpotential of HER at −10 mA cm^−2^ was 21 mV (Figure 6f). The overpotential of OER at 10 mA cm^−2^ was 192 mV (Figure 6g). Because of Ir doping in the LDH lattice, the synthesized NiCo LDH/NF has better durability than the Ir-loaded NiCo LDH and sustainable stability of more than 200 h. The overall water decomposition performance (Figure 6h) of NiCo LDH/NF was obtained in 1.0 M KOH electrolyte at 10 mA cm^−2^ at 1.45 V low battery voltage. The remarkable properties of Ir-NiCo LDH can be attributed to the doping of Ir in LDH lattice, which not only promotes their synergistic effect in HER/OER process but also provides a large electrochemical active surface area and improved conductivity.

### 3.3. Electrocatalysts of LDHs Supporting Ru

It was found that ruthenium (Ru) is also an excellent material for the design of bifunctional electrocatalysts [93]. So far, numerous studies have shown that Ru and RuO_2_ exhibit excellent performance in HER and OER [94]. It should be noted that when the applied potential is greater than 0.04 V (vs. RHE), the sub-nano Ru is easily oxidized to RuO_2_, which provides effective guidance for the development of bifunctional catalysts with good properties for HER and OER [73]. Considering the low dosage caused by large grain size, it is imperative to reduce the size of Ru. Recently, Xi et al. [95] developed a Ru-doped NiFe-based catalyst for three-dimensional nanoporous surfaces. In situ generated metal (hydrogen) oxides and nano-porous structures provide a rich active center. The overpotential is 245 mV at 10 mA cm^−2^, the slope of Tafel is 15 mV dec^−1^ and it has excellent OER performance. However, pushing the catalysts to sub-nanoscale is not easy because they are thermodynamically unstable and tend to clump together. It is an effective strategy to improve the performance of HER and OER at the same time by building Ru sub-nanoclusters on porous NiFe-LDH through electron coupling and synergetic effect [96].

Chen et al. [67] proposed a new strategy to use Ru to partially replace Fe atoms to greatly accelerate the rate of hydrogen evolution of NiFe-LDH. They used hydrothermal reaction to grow Ru-doped NiFe-LDH nanosheets (Figure 7a). Many nanosheets with a size of 90–180 nm are grown on the underlying nickel foam; they grow vertically and are connected to each other (Figure 7b). A large number of highly dispersed, brightly contrasted Ru atoms are present on NiFeRu-LDH nanoplates with a plane spacing of 0.25 nm, corresponding to the (012) plane of NiFeRu-LDH (Figure 7c,d). It is noteworthy that the synthesized Ru-doped NiFe-LDH nanosheet (NiFeRu-LDH) displays good HER performance. At the current density of 10 mA cm^−2^, the overpotential is only 29 mV, lower than the precious metal Pt/C catalyst (31 mV at 10 mA cm^−2^) (Figure 7e). The bifunctional NiFeRu-LDH electrocatalyst was used as anode and cathode in the total hydrolysis reaction, and the alkaline cell with current density of 10 mA cm^−2,^ could be driven stably at very low 1.52 V battery voltage. The cell voltage is lower than that of the Pt/C-Ir/C couple (1.60 V at 10 mA cm^−2^) (Figure 7f). Both the experiment and density functional theory (DFT) calculation results show that the introduction of Ru atoms in NiFe-LDH can effectively reduce the energy barrier of the Volmer step and ultimately accelerate the HER dynamics under alkaline conditions (Figure 7g). This solution not only provides a new way to replace platinum catalysts for the preparation of HER but also opens up a new way for the development of low-cost, high-activity electrocatalysts for other catalytic reactions related to energy conversion.

Wang et al. [96] reported the preparation of an efficient NiFe-LDH supported nano-Ru bifunctional electrocatalyst by two-step method (Figure 8a). The adsorption energy of H* can be increased and the adsorption kinetics can be improved by combining sub-nano Ru with NiFe LDH. The results show that Ru/NiFe LDH-F/NF nanoarrays have good OER and HER properties. The overpotential of OER and HER at current density of 10 mA cm^−2^ was 230.0 mV and 115.6 mV (Figure 8b,c). Ru/NiFe LDH-F/NF was used as HER and OER electrodes to prepare a 10 mA cm^−2^ total electrolyzed water with a potential of 1.53 V (Figure 8d). In addition, theoretical calculations show that the adsorption energies of H* and OH* can be optimized at the Ru-NiFe LDH interface (Figure 8e,f). This enhancement can be attributed to the unique multi-dimensional structure between Ru and NiFe LDHs, as well as the intermediate modulation triggered by electron coupling and synergy effects. This study provides new insights for the development of high efficient bifunctional electrocatalysts for water electrolysis.

### 3.4. Electrocatalysts of LDHs Supporting Other Noble Metals

The noble metals (Pt, Ir and Ru) described above have been extensively studied for their excellent catalytic activity as active sites of water electrolysis catalysts [73]. Furthermore, it is worth mentioning that besides the above mentioned precious metals, other precious metals (such as Au, Pd and Rh) also have high catalytic activity as electrocatalysts [97,98]. For example, Taei et al. [99] have successfully prepared AuNPs@CaFe-LDH composite on CaFe-LDH surface by electrodeposition and used it as an efficient electrocatalyst for OER and HER in alkaline solution. Compared with AuNPs, CaFe-LDH and CaFe-LDH@AuNPs catalysts, AuNPs@CaFe-LDH catalysts have significantly preferable OER and HER performance. Meanwhile, AuNPs@CaFe-LDH catalyst has excellent stability. The high activity of AuNPs@CaFe-LDH is related to the synergistic effect of AuNPs with CaFe-LDH and the higher electrochemical surface area provided by AuNPs. Further investigation of the surface materials of the catalysts will provide useful information for the design of better OER and HER electrocatalysts.

Compared with noble metals such as Pt and Ir, the research on the combination of Pd and non-noble metal catalysts to improve the electrocatalytic activity, especially the electrocatalytic activity of hydroxide, is still at the initial stage, but there are some related studies. Guo et al. [80] used a hydrothermal method to prepare layered NiFe LDH on a foamed nickel substrate and prepared ultra-fine Pd nanoparticles by electrodeposition to achieve improved bifunctionality of electrocatalysts. The introduction of palladium can induce more active centers, strong electrical interactions, and enhance charge transfer, resulting in a significant increase in the catalytic activity of water electrolysis. The Pd-NiFe LDH exhibited impressive catalytic activity. Under the condition of current density of 10 mA cm^−2^, OER and HER exhibited 156 mV and 130 mV, respectively. The dual-electrode electrolyzer assembled with Pd-NiFe LDH can achieve 1.514 V ultra-low battery potential water splitting reaction under the condition of 10 mA cm^−2^.

Other researchers have found that Rh initially combines with NiFe-LDH in the form of oxide dopant and metal clusters to significantly improve HER kinetics without sacrificing OER properties. Zhang et al. [100] demonstrated that the combination of Rh and NiFe-LDH can significantly improve HER dynamics performance without sacrificing OER properties. In this study, Rh-loaded Ru/NiFeRu-LDH was directly grown on nickel foam by a simple hydrothermal method. Rh was initially combined with NiFe-LDH in the form of oxide dopant and metal clusters. In the synthetic materials, some of Rh replaced the iron centers in the NiFe-LDH, and some of them were in the form of metal clusters (<1 nm). The Rh/NiFeRh-LDH can catalyze HER current density of 10 mA cm^−2^ with only 58 mV over-potential in 1 M KOH electrolyte and the current density of 10~190 mA cm^−2^ with only 1.46~1.7 V battery voltage (Figure 9a,b). This performance is 5–6 times higher than the reference cell of 20% Pt/C || RuO_2_ electrode and better than the most advanced bifunctional electrocatalyst (Figure 9c,d).

## 4. Conclusions and Prospectives

Electrolytic water reaction is considered to be an ideal method to produce carbon-free high-energy hydrogen fuel. LDHs materials have been widely studied in the electrocatalysis field, due to their unique layered structure, tunable metal species and highly dispersed active sites. In this review, the research progress of LDHs supporting noble metal electrocatalysts in recent years is reviewed; the applications of LDHs supporting noble metal catalysts (Pt, Ru, Ir, etc.) in electrocatalysis were briefly introduced from the aspects of preparation and modification. By using the highly specific surface area and porous structure of LDHs, the size and dispersibility of noble metal nanoparticles can be controlled, and the agglomeration of noble metal nanoparticles can be prevented effectively; thus, the high utilization ratio of noble metal nanoparticles can be realized. Moreover, the LDHs supporting noble metal catalysts obtained through the topotactic transformation of LDHs precursors significantly reduced the energy barrier of electrolyzing water, showing remarkable catalytic activity, good conductivity and excellent durability. Despite these advances, there are still a number of issues and challenges to be explored to improve their electrocatalytic performance and cost-effectiveness. In order to achieve the goal of hydrogen fuel application, we propose several directions for future research work:

(i) Accurate control of the structure of LDHs is essential, and effective and advanced synthesis methods can be used to manipulate its composition, morphology, size, interface and nanostructure, for example, LDH nanoarrays with more uniform arrangement, layered porous LDH nanosheets and ultrathin monolayer LDHs.

(ii) By adjusting the size of noble metal nanoparticles and dispersing them evenly on the support, the amount of noble metal can be reduced to the maximum, and the electrocatalytic performance can be improved at low cost. The more important goal is to replace rare and precious metals with cheaper and richer metals, making the process greener and greener and more cost-effective.

(iii) The electronic structure or conductivity can be adjusted effectively by introducing dopant or defect. By adjusting the electronic structure, the conductivity and intrinsic catalytic activity of the active site can be increased, the charge transfer can be promoted and the catalytic activity can be improved.

(iv) It is very important to conduct the theoretical calculation of materials. The theoretical calculation is helpful to estimate the adsorption and desorption capacity, band gap and free energy change of each catalytic step, and to provide theoretical support for researchers to select LDH materials with excellent properties.

(v) In order to meet the practical needs of large-scale application, the performance and durability under a high current should be paid more attention. The reasons for the reduction of catalytic activity, such as electrolyte corrosion and catalyst shedding, should be systematically studied, in order to guarantee that the actual production has good stability.

The investigation of practical LDHs-based electrocatalysts has become a research hotspot in the field of electrolyzed water. With the joint efforts of scientists from all over the world, we believe that LDHs as promising electrocatalysts will be properly developed and make a significant contribution to the industrial utilization of hydrogen fuel.

## Data Availability

Not applicable.

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
