# Peer review of "Recent Advances in Layered-Double-Hydroxides Based Noble Metal Nanoparticles Efficient Electrocatalysts"

_nanomaterials, 2021, doi:10.3390/nano11102644_

Round 1

Reviewer 1 Report

Manuscript «Recent Advances in Layered Double Hydroxides Based Noble 2 Metal Nanoparticles Efficient Electrocatalysts” by Z. Zhang et al.

The authors propose a comprehensive review of recent work on LDH supporting noble metal electrocatalysts. The field of research is very timely and relevant in the framework of efficient hydrogen production, which is a major limiting factor in the development of the use of hydrogen as energy vector. The review is well written and considers a large number of recent and complementary works, the theoretical approach remains modestly addressed. The illustrations are overall well chosen and adequate. The paper will be very useful for researchers working in - or starting to work in - the field and clearly deserves publication in my opinion.

I found only a few points that the authors may consider:

  • Line 63. “maximize” may not be well chosen here, perhaps “optimize” ?
  • Line 77. Maybe “cation” instead of “cations”
  • Line 78. Is maybe “to” missing between “leading” and “high”?
  • Caption figure 4 refers 2 times to Figure 1 instead of Figure 4, please correct.
  • Figure 6 is too dense/small and is poorly readable. I would suggest to put panels (g) (i) and (k) in an additional row instead of a column to make to whole figure more readable.

Author Response

Reviewer #1: The authors propose a comprehensive review of recent work on LDH supporting noble metal electrocatalysts. The field of research is very timely and relevant in the framework of efficient hydrogen production, which is a major limiting factor in the development of the use of hydrogen as energy vector. The review is well written and considers a large number of recent and complementary works, the theoretical approach remains modestly addressed. The illustrations are overall well chosen and adequate. The paper will be very useful for researchers working in - or starting to work in - the field and clearly deserves publication in my opinion.

Response : Thank you very much for your encouragement to our work and for your valuable comments. We are encouraged by your comments.

I found only a few points that the authors may consider:

Point 1: Line 63. “maximize” may not be well chosen here, perhaps “optimize” ?

Response : Thanks for the valuable comments. We have replaced “maximize” with “optimize”, the corresponding sentence is revised as follows:

Thus, researchers begin to focus to explore effective and efficient strategies to achieve high dispersion of active substances, so as to optimize the use of noble metal resources.

(On Page 2, lines 61-63 of the revised manuscript)

Point 2: Line 77. Maybe “cation” instead of “cations”

Response : Many thanks! We have replaced “cation” with “cations”, which reads as follows:

Each single metal cation with catalytic activity can be used as a single catalytic active site, leading high catalytic activity in the electrocatalysis process. (On Page 2, lines 76-78 of the revised manuscript)

Point 3: Line 78. Is maybe “to” missing between “leading” and “high”?

Response : Thanks for the valuable comments. We have revised the content as follows:

Each single metal cation with catalytic activity can be used as a single catalytic active site, leading to high catalytic activity in the electrocatalysis process. (On Page 2, lines 76-78 of the revised manuscript)

Point 4: Caption figure 4 refers 2 times to Figure 1 instead of Figure 4, please correct.

Response : Many thanks! We have revised the content as follows:

Fig.4 (a) HRTEM images of Pt-NiFe LDH nanosheet. The illustration in Figure 4a shows the size distribution of Pt sub-nano clusters. (b) LSV curves of various electrodes at 1 mV s-1 scanning speed. (c) LSV curves of the Pt-NiFe LDH/CC electrode before and after 1000 cycles. (d) CP curve of the Pt-NiFe LDH/CC electrode lasting 36000 s at a constant current density of 10 mA cm-2. (On Page 7, lines 214-221 of the revised manuscript)

Point 5: Figure 6 is too dense/small and is poorly readable. I would suggest to put panels (g) (i) and (k) in an additional row instead of a column to make to whole figure more readable.

Response : Thanks for the valuable comments. Another reviewer also suggested Figure 6 need to be modifiedin the original manuscript. We have modified Figure 6 in the revised manuscript. We removed Figure 6g, Figure 6i and Figure 6k from the original manuscript. The revised image is as shown in Figure R1:

Figure R1 (a) The simple preparation flow chart of Ir-NiCo LDH/NF. (b)-(c) SEM and TEM image of Ir-NiCo LDH/NF. (d)-(e) High-resolution TEM and high-resolution HAADF-STEM image of Ir-NiCo LDH/NF. (f) Polarization curves of Ir-NiCo LDH and its comparison samples in N2-saturated 1.0 M KOH for HER. (g) Polarization curves of Ir-NiCo LDH and its comparison samples in N2-saturated 1.0 M KOH for OER. (h) Polarization curves in N2-saturated 1.0 M KOH for overall water splitting.

(On Page 9 of the revised manuscript)

Reviewer 2 Report

This paper demonstrated a review for the noble metal nanoparticles modified LDHs. The LDH supporting Pt, Ir, Ru and other noble metal NP was introduced as electrocatalysts for water splitting reactions such as OER and HER. The authors emphasized the recent advance of LDHs supporting noble metal catalysts, however, the LDH are somewhat limited to Ni based ones such as Ni/Fe or Ni/Co and etc. 

Frist, the introduction which explained the energy crisis and requirement of new water splitting catalyst seems excellent guide line for readers. Then, the main content is simple but describes well from the synthesis to the catalytic activities of the electrocatalysts. Finally, the author offers the challenges and outlooks in conclusion part. Overall, it is well constructed.

Compared to the description of various noble metal nanoparticles, LDH was mainly explained with Ni-base ones.

Is there a special reason?

If it has been mainly reported that Ni based (or NiFe) LDH has been used recently, it would be better to mention the recent trend of the LDH.

Author Response

Thank you very much for your encouragement as well as for the affirmation and support of our manuscript. We have carefully responded to your valuable comments and revised them. Through the form of attachments sent to you, please check the receipt. Thank you again for your affirmation and support of our work. Wish you a pleasant work.

Reviewer 3 Report

The topic covered in this review is one of great importance and the authors have done well to write this manuscript describing the state of the art in this field.
the review presents a very well organized graphic apparatus, which presents with clarity and completeness the various points discussed in the manuscript. 
the manuscript reads with pleasure and I have no observations or criticisms to make of the authors. Therefore it is with pleasure that I suggest the publication of this manuscript. 
There is only one point that I would like to bring to the attention of the authors 
Figures 4,5 and 6 which report some results taken from the literature are very rich of information, much more than what the authors discuss in their review. I suggest to eliminate the results not directly considered and commented in this manuscript, or to add a comment in order to make more complete the discussion.
Finally, but just to be picky, I would expand the conclusions at the end of the manuscript a bit.
However these are small things, mostly related to personal taste rather than being real flaws in this work. I would suggest that the authors themselves decide whether or not to take them into account. 

Author Response

(The authors gave the same response as above.)
